# A Simple and Debiased Sampling Method for Personalized Ranking

## Abstract

Pairwise ranking models have been widely used to address various problems, such as recommendation. The basic idea is to learn the rank of users' preferred items through separating items into *positive* samples if user-item interactions exist, and *negative* samples otherwise. Due to the limited number of observed interactions, pairwise ranking models face serious *class-imbalance* issue. Our theoretical analysis shows that current sampling-based methods cause the vertex-level imbalance problem, which makes the norm of learned item embeddings towards infinite after a certain training iterations, and consequently results in vanishing gradient and affects the model performance. To this end, we propose VINS, an efficient *Vital Negative Sampler*, to alleviate the class-imbalance issue for pairwise ranking models optimized by gradient methods. The core of VINS is a bias sampler with reject probability that will tend to accept a negative candidate with a larger popularity than the given positive item. Evaluation results on several real datasets demonstrate that the proposed sampling method speeds up the training procedure 30% to 50% for ranking models ranging from shallow to deep, while maintaining and even improving the quality of ranking results in top-N item recommendation.

## 1 Introduction

Offering personalized service to users is outstanding as an important task, for example, ranking the top-$N$ items that a user may like. Solutions to such kind of problems are usually designed on a bipartite graph $G = (V, E)$, where vertex set $V = U \cup I$ contains user set $U$ and item set $I$, and $E$ denotes the edge set. Each edge $e_{ui} \in E$ denotes an observed interaction between user $u$ and item $i$. Users' preference on items is modeled by *pairwise loss* functions with the assumption that items with interactions from a user are of more interest to this user than those without interactions. The loss function thus involves pairwise comparison between an observed (*positive*) edge $e_{ui} \in E$ and an unobserved (*negative*) edge $e_{uj} \notin E$. The optimization process thus suffers from the *class-imbalance* issue ,because in practical scenario, the number of observed (*positive*) edges are always much less than the unobserved (*negative*) ones. The imbalance between $e_{ui} \in E$ and $e_{uj} \notin E$ can be regarded as the *edge-level* imbalance issue.

Pioneering works dealing with the class-imbalance problem can be categorized into two main families: using *stationary* sampling or using *dynamic* sampling. Approaches in the former family usually start from the *edge-level* class-imbalance issue through under-sampling negative edges from a pre-defined stationary distribution (Rendle et al., 2009; Rendle and Freudenthaler, 2014), or over-sampling positive edges by creating instances through the social connection (Chen et al., 2019). However, they ignore that class-imbalance issue also exists in vertex side because each vertex can appear in both positive and negative edges. Through some basic statistical analysis, we acquire some interesting findings, that is, the vertex degree has positive impact on *vertex-level imbalance* problem. If we sample negative instances from a stationary distribution, those popularity vertexes with degree greater than average vertex degree are under-sampled as negative samples, while "cold-start" vertexes with degree less than average degree are over-sampled. Moreover, they can't capture the dynamics of relative ranking order between positive and negative samples, as shown in Figure 1(a) and 1(b). From Figure 1(a) we can see that it's easy to find an order-violated item for pairwise loss optimization at the initial state, because there are many negative items ranking higher than the positive item. However, as the learning process moves forward, massive number of negative items are distinguished well from the positive item, shown in Figure 1(b). At this time, a large portion of

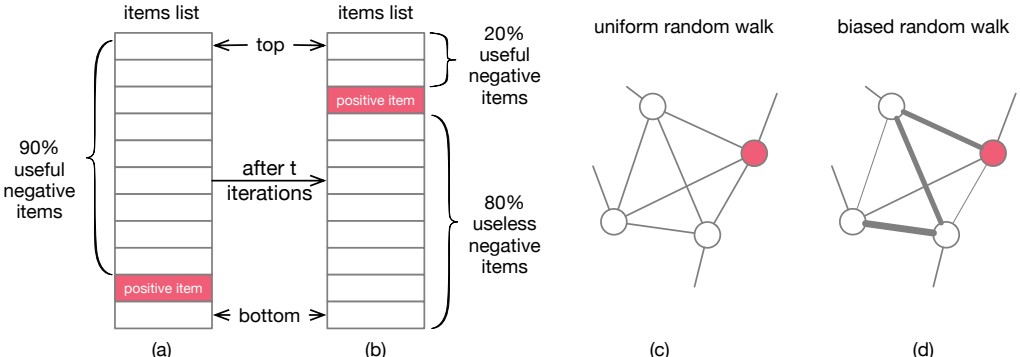

Figure 1: Illustration of finding useful negative items for pairwise loss optimization: (a) is the initial stage of optimization when it's easy to get one negative item; (b) shows that useful negative items are more difficult to get as the learning process moves forwards; (c) sampling negative items from uniform distribution equals to do unbiased random walk on fully connected item-item graph; (d) presents an alternative solution depending on a bias random walk.

the negative items are useless for pairwise loss optimization, because they already rank lower than the positive item.

Recently dynamic sampling approaches (Weston et al., 2011; Yuan et al., 2016; Wang et al., 2020) have shown their significant contribution to negative instances selection by considering the hardness of sampling a negative sample. However, existing dynamic methods have several drawbacks: 1) they lack systematically understanding their connection to class-imbalance issue, leading to only sampling candidate from uniform distribution; 2) they have to find a violated negative sample through searching massive candidates, causing high computation complexity (over ten times higher than sampling from stationary distribution).

In this work, we aim at finding clues that can help to design a **faster dynamic negative sampler** for the personalized ranking task. We find that sampling from uniform distribution can be regarded as a random walk with a transition probability matrix $\mathcal{P}$ for arbitrary node pair in a fully connected item-item graph, which is presented in Figure 1(c). Intuitively, nodes (items) are different in their nature (e.g., degree, betweenness). A biased transition matrix $\mathcal{P}^*$ might be more helpful on finding the desired negative items, than a uniform random $\mathcal{P}$, as shown in Figure 1(d). Through theoretical analysis, we find that one of the potential solutions to decode the biased transition process and walking with a biased transition matrix $\mathcal{P}^*$ is to tackle the class-imbalance issue. To achieve this goal, it is essential to first dissect the impact of class-imbalance issue. More specifically, we mainly investigate the three questions: Q1) how the class-imbalance problem is reflected in current sampling-based pairwise ranking approaches? Q2) what is the impact of the imbalance problem on learning optimal pairwise ranking model? Q3) how can we resolve the class-imbalance issue and design a faster dynamic sampling approach to boost ranking quality? We answer the above questions with theoretical analysis in Section 3. The brief summary is, to Q1, if negative instances are sampled from a uniform distribution (e.g., in (Rendle et al., 2009)), vertexes with high degrees are under-sampled as negative samples, while "cold-start" vertexes with low degrees are over-sampled. To Q2, we theoretically show that the class-imbalance issue will result in frequency gathering phenomenon where the learned embeddings of items with close popularity will gather together, and cause gradient vanishment at the output loss. Based on the above insights, for Q3, we propose an efficient *Vital Negative Sampler* (VINS), which explicitly considers both *edge-* and *vertex-*level class-imbalance issue. In summary, our contributions of this work are as follows:

- We indicate out *edge-* and *vertex-level* imbalance problem raised in pairwise learning loss, and provide theoretical analysis that the imbalance issue could lead to frequency gathering phenomenon and vanishing gradient at the output loss.

- To address the class-imbalance and vanishing gradient problem, we design an adaptive negative sampling method with a reject probability based on items' degree differences.

- Thoroughly experimental results demonstrate that the proposed method can speed up the training procedure 30% to 50% for shallow and deep ranking models, compared with the state-of-the-art dynamic sampling methods.

## 2 RELATED WORK

Pairwise comparison usually happens between an observed (*positive*) and an unobserved (*negative*) edge, when the interactions between users and items are represented as a bipartite graph. Such an idea results in a serious *class-imbalance* issue due to the pairwise comparison between a small set of interacted items (*positive as minority class*) and a very large set of all remaining items (*negative as majority class*). Pioneering work proposed in (Rendle et al., 2009) presented an under-sampling approach via uniformly sampling a negative edge for a given positive edge. Following the idea in (Rendle et al., 2009), (Zhao et al., 2014) proposed an over-sampling method by employing social theory to create synthetic positive instances. (Ding et al., 2019) augmented pairwise samples with view data. However, these sampling strategies discard a fact that each item has its own properties, *e.g.*, degree, betweenness. (Rendle and Freudenthaler, 2014) considered vertex properties and proposed to sample a negative instance from an exponential function over the order of vertex degree. Despite of the effectiveness and efficiency of sampling from a stationary distribution (e.g., uniform, or power function over vertex popularity), they ignore the impact of relative order between positive and negative samples during the learning processes, as shown in Figure 1(a) and 1(b).

Recently dynamic sampling approaches (Weston et al., 2011; Yuan et al., 2016; Chen et al., 2018) aiming at estimating the rank order of positive samples have shown significant contribution of selecting vital negative instances. As a pioneering work, (Weston et al., 2011) proposed the WARP loss aiming at playing less attention to well-learned positives, but more emphasis on the low-rank ones. However, along with the growing of iterations, sampling a violated negative items become very difficult (Hsiao et al., 2014). WARP inspires lots of recent works to estimate rank-aware weight from a uniform distribution As the state-of-the-art variant of WARP loss, LFM-W (Yuan et al., 2016) advances WARP with a normalization term. However, estimating the rank-aware weight from a uniform distribution makes LFM-W need lots of steps to find a violated sample. Moreover, LFM-W might find sub-optimal negative sample without considering the class-imbalance issue. Besides considering ranking order, (Wang et al., 2019) regarded dynamic sampling as a minmax game. VINS also inherits the basic ideas from WARP but modifies the target distribution and proposes to estimate it through an importance sampling method after theoretically investigating the existing class-imbalance issue and its potential influence. LFM-W can be regarded as a special case of the proposed VINS with a proper setting.

## 3 CLASS IMBALANCED ANALYSIS

Let's use $G = (V, E)$ to represent a user-item interaction graph, where vertex set $V = U \cup I$ contains users $U$ and items $I$, and $e_{ui} \in E$ denotes an observed interaction (*e.g.* click, purchase behaviors) between user $u$ and item $i$. The relationship between user $u$ and item $i$ can be measured by a factorization focused method, known as $x_{ui} = P_u \cdot P_i$, where $P_u = f(u|\theta_u) \in \mathbb{R}^d$ and $P_i = g(i|\theta_i) \in \mathbb{R}^d$ are the representation of user $u$ and item $i$ generated by deep neural network $f(\cdot)$ and $g(\cdot)$ with parameters $\theta_u$ and $\theta_i$, respectively. To learn vertex representation that can be used to accurately infer users' preferences on items, pairwise ranking approaches usually regard the observed edges $e_{ui}$ as positive pairs, and all the other combinations $e_{uj} \in (U \times I \setminus E)$ as negative ones. Then a set of triplets $D = \{(u, i, j)|e_{ui} \in E, e_{uj} \in (U \times I \setminus E)\}$ can be constructed base on a general assumption that the induced relevance of an observed user-item pair should be larger than the unobserved one, that is, $x_{ui} > x_{uj}$. To model such contrastive relation, one popular solution is to induce pairwise loss function as follows:

$$\mathcal{L}(G) = \sum_{(u,i,j)\in D} w_{ui} \cdot \ell_{ij}^u(x_{ui}, x_{uj}), \tag{1}$$

where $\ell_{ij}^u(\cdot)$ can be hinge, logistic or cross entropy function that raises an effective loss for any triplet with incorrect prediction (*i.e.* $x_{uj} > x_{ui}$) that violates the pairwise assumption. $w_{ui}$ is the a weight factor which shows the complexity to discriminate the given comparison sample. The optimization of Equation (1) involves an extreme class-imbalance issue, because in practical scenario, the number of unobserved interactions $e_{uj} \notin E$ (negative) is usually extremely larger than the observed $e_{ui} \in E$ (positive). The imbalance between $e_{ui} \in E$ and $e_{uj} \notin E$ in pairwise loss can be regarded as the ***edge-level imbalance*** issue. Since the class-imbalance problem is caused by the majority of negative edges, under-sampling majority is a practical solution for it (Rendle et al., 2009; Mikolov et al.,

2013). Let's take the most popular strategy of under-sampling negative edges as an example. For a given positive edge $e_{ui} \in E$, we can sample a negative edge by fixing user $u \in U$, then sample one item $j \in I, e_{uj} \notin E$ with replacement from a static distribution $\boldsymbol{\pi} = \{\pi(i), i \in I\}$, where $\pi(i) = d_i^\beta, \beta \in [0,1]$ denotes a weight function of item degree $d_i$. Then we can optimize the objective function in Equation (1) with the constructed pairwise samples $\tilde{D} \in D$. **In most of pairwise ranking models, how to select effective pairwise comparison samples plays an indispensable role in boosting the ranking performance**. In the following, we'd like to present the challenges raised by the class-imbalance issue on selecting pairwise comparison samples, and how to address these challenges with an adaptive sampling method.

### 3.1 VERTEX-LEVEL IMBALANCE FROM SAMPLING (Q1)

Under-sampling approach can well solve the edge-level imbalance issue. However, it will introduce ***vertex-level imbalance*** issue, which has not been aware of, and initiates our study.

**Definition 3.1** (Vertex-level Imbalance). *A vertex can appear in either positive or negative edges. In our case, item $i$ appears as a positive one for user $u$, but can be a negative one for other users. Vertex-level imbalance happens when the number of times that a vertex appears in observed edges is extremely smaller or larger than that in the unobserved ones.*

Assuming that in each iteration of optimizing Equation (1), we will sample one negative edge for each observed edge $e_{ui}$. With a given graph $G$ with $|E|$ observed edges, item $i$ can only appear in $d_i$ edges as positive samples. In other words, item $i$ could appear as negative in the other $|E| - d_i$ edges with probability $p(i)$ when sampling with a static distribution $\boldsymbol{\pi}$ defined as $p(i) = \pi(i)/\sum_{j \in I} \pi(j)$. Then, the expected number of times that the item $i$ acts as a negative sample is $p(i) \cdot (|E| - d_i)$. Afterwards, we define the imbalance value ($IV$) of item $i$ as: $IV(i) = \frac{d_i}{p(i) \cdot (|E| - d_i)} = \frac{d_i^{1-\beta} \cdot \sum_{j \in I} \pi(j)}{|E| - d_i}$. Through theoretical analysis, we find that imbalance value is positively correlated to item degree.

**Theorem 3.1.** *By sampling negative items with a static distribution $\boldsymbol{\pi} = \{\pi(i) = d_i^\beta | \beta \in [0,1], i \in I\}$, for two items with $d_i > d_j$, the imbalance value of item $i$ is larger than item $j$.*

The complete proof for theorem 3.1 can be found in Appendix A.

The above analysis shows that the degree of the most popular and long-tailed item will determine the upper and lower bound of item imbalance value for a given graph $G$. As a special case, if $\beta = 0$, we have $IV(i) = \frac{d_i \cdot |I|}{|E| - d_i}$. Let's set $IV(i) = 1$, we can see that $d_i = \frac{|E|}{|I|+1} \approx \frac{|E|}{|I|}$, which is exact the average item degree. If an item's degree is larger than the average degree, it will have an imbalance value larger than 1, while for those item with degree lower than average degree, their imbalance value will be smaller than 1. This implies that popular vertexes are under-sampled as negative samples, while "cold-start" vertexes are over-sampled. For different setting of $\beta$, the situation will be different. We illustrate the maximum and minimum imbalance value in Figure 2, obtained by the empirically calculated $IV(i)$ from two real datasets with different decay factor $\beta$. We can see that a proper choice of decay factor $\beta$ can reduce the maximum imbalance, meanwhile increase the minimum value.

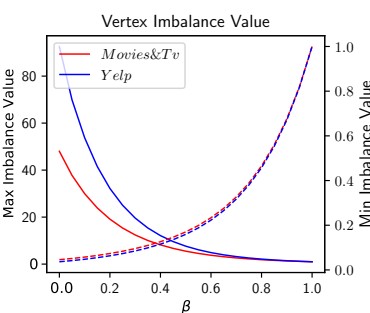

Figure 2: Maximum (solid line) and minimum (dash line) imbalance value along with different decay parameter $\beta$ on Yelp and Amazon Movies&Tv datasets.

### 3.2 IMPACT OF CLASS-IMBALANCE (Q2)

We next move to the question "**what is the impact of the class-imbalance problem on pairwise loss function optimization?**". Before answering this question, we first introduce an imbalanced item theorem inspired by the *Popular Item Theorem* (Lee and Lin, 2016), which proves that the norm of latent vector of the popular items will be towards infinite after a certain number of iterations. We extend the theorem as follows (with complete proof presented in the Appendix B):

**Theorem 3.2.** *[Imbalanced Item Theorem] Suppose there exists an imbalanced item $i$ with $IV(i) \gg 1$, such that for all neighbor users $u \in \mathcal{N}_i$, $x_{ui} \geq x_{uj}$ for all other observed item $j$ of user $u$. Furthermore, after certain iterations $\tau$, the representation $P_u, u \in \mathcal{N}_i$ converges to certain extent. That is, there exists a vector $\hat{P}^t$ in all iteration $t > \tau$, inner-product $(\hat{P}^t, P_u^\tau) > 0$. Then the norm of $P_i$ of the imbalanced item $i$ will tend to grow to infinity if $\frac{\partial \ell_{ij}^u}{\partial x_{ui}} > 0$ for all $i$ with $x_{ui} > x_{uj}$.*

**Frequency gathering phenomenon.** The *Imbalanced Item Theorem* implies that the learned embeddings of items will appear a certain pattern that is closely related to item's imbalance value. To confirm that, we optimize logistic pairwise loss function by sampling negative samples from a uniform distribution and also by using the proposed method VINS on the experimental data. Since there's no vertex-level imbalance problem in the user side, the learned user embeddings are independent on the degree information. From Figure 3, we can see that the learned embeddings of items by the uniform sampling approach appear clear *frequency gathering* phenomenon, where items with similar degree values gather together. The more popular items tend to have larger embedding norms, which matches the statement in the *Imbalanced Item Theorem*. We further study the embeddings learned by the proposed approach VINS that explicitly considers vertex-level class-imbalance, and find that those bottom items tend to spread across the frequency margins. We also explore a special case $\beta = 1$ for sampling from a static distribution $\boldsymbol{\pi}$, with which the vertex-level imbalance can be perfectly mitigated. However, comparing with the setting $\beta = 0$, it can significantly downgrade the prediction performance, because the learned item embeddings can't keep their structure role and proximity very well. It suggests that controlling the class-imbalance problem might help to improve the ranking performance, but still need to achieve a trade-off between keeping the graph structures and alleviating the negative impact of class-imbalance problem.

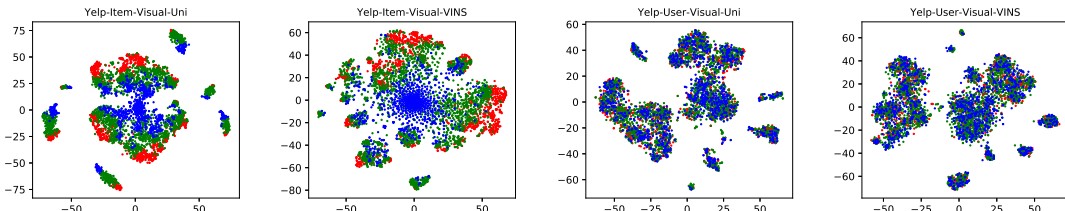

Figure 3: Visualizing the projection of learned embeddings with the classical uniform sampling and the proposed sampler *VINS* by the T-SNE algorithm into two-dimensional space (colored by vertex degree levels, red–top 25%, blue–bottom 25%, green–the rest).

**Gradient Vanishment.** Besides the frequency gathering phenomenon, another issue caused by the infinite norm is the gradient vanishment in pairwise loss optimization. Following the under-sampling method described in Section 3, gradient update for model parameters can be carried out for a given pairwise sample $(u, i, j)$. After $t > \tau$ iterations, the model parameters $\theta_i$ can be updated with stochastic gradient descent method: $\theta_i^{t+1} = \theta_i^t + \eta \cdot \lambda_{ij}^u \cdot \frac{\partial x_{ui}}{\partial \theta_i}$, where $\lambda_{ij}^u = \frac{\partial \ell_{ij}^u}{\partial x_{ui}}$, and $\eta$ denotes the learning rate, and $\frac{\partial x_{ui}}{\partial \theta_i}$ represents a gradient backpropagation operation according to the chain rule.

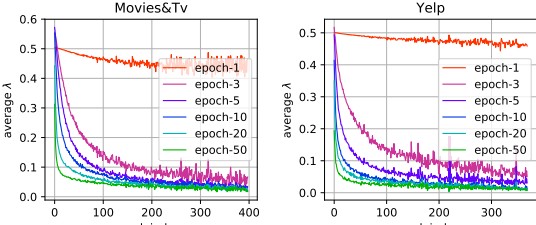

Figure 4: Illustration to show the connection between class-imbalance and gradient vanishment. The rank index stands for the ranking position of imbalance value (IV) in ascending order. We use average gradient magnitude $\lambda$ as the *y*-axis.

The value of $\lambda_{ij}^u$ depends on the type of loss function. If we use logistic loss as an instance, $\ell_{ij}^u(x_{ui}, x_{uj}) = \ln \sigma(x_{ui} - x_{uj})$, where $\sigma(x)$ is the sigmoid function and $\lambda_{ij}^u = (1 - \sigma(x_{ui} - x_{uj}))$. According the *Imbalanced Item Theorem*, the norm of learned embeddings of those imbalanced items will become extremely large. Let's fold out $x_{ui} = P_u \cdot P_i = ||P_u|| \cdot ||P_i|| \cdot cos(P_u, P_i)$. If positive item $i$ suffers from imbalanced issue and has a large norm, i.e., $||P_i|| \gg ||P_j||$, the relevance prediction for user $u$ will be dominated by the norm of item $i$'s embedding. Then, the induced hinge loss will be very close to zero. While popular items take up a large portion of the observed edges, most of the training samples will have $\lambda_{ij}^u \to 0$ according to Theorem 3.1 and Theorem 3.2. **It suggests that massive number of pairwise samples are meaningless for updating the model, and only a small number of them are valuable.** Following the idea in pioneering work (Rendle and

Freudenthaler, 2014), we conduct an empirical study on two experimental data to show the connection between gradient vanishment and item class-imbalance issue. From the results shown in Figure 4, we have the same observation as (Rendle and Freudenthaler, 2014) that gradient magnitude of most of training cases tend to be close to zero, meanwhile we find that items with larger imbalance value tend to have smaller gradient magnitude on average. Connecting to the Theorem 3.1 and Theorem 3.2, we can see that the reason why popular items tend to have smaller gradient magnitude has positive relation to the class-imbalance issue.

## 4 VITAL NEGATIVE SAMPLER

In this section, we will introduce the proposed method, namely Vital Negative Sampler (VINS). We first introduce RejectSampler which is the key component of VINS, then present VINS.

### 4.1 SAMPLING WITH REJECT PROBABILITY (REJECTSAMPLER)

Combining Theorem 3.1 and the frequency gathering phenomenon, **we find that there exists a positive connection between item degree and the learned embeddings**. We thus design a negative sampling approach which tends to sample a negative item $j$ with a larger degree than the positive item $i$, rather than a negative item with a smaller degree than item $i$. With such strategy, it's helpful to control the class-imbalance issue by reducing the imbalance value of popular items, but increase the imbalance value of long-tailed items. More specifically, for a given positive sample $e_{ui}$, we sample a negative item $j$ with reject probability 1 - $min\{\frac{\pi(j)}{\pi(i)}, 1\}$. With this reject probability, we can increase the chances of popular items exposed as negative samples while downgrading the chances of long-tailed items. We can see that the RejectSampler actually equals to a biased random walk as shown in Figure 1(d) to choose the next step with a given transition matrix $P^*$, where

$$P_{ij}^* = \begin{cases} min\{\frac{\pi(j)}{\pi(i)}, 1\} & if \quad i \neq j \\ 1 - \sum_{v \neq i} \mathcal{P}_{iv}^* & if \quad i = j \end{cases}$$ . In fact, *RejectSampler* can adapt beyond the item degree

information to define the reject probability, resulting a different transition matrix $\mathcal{P}^*$. The detail of *RejectSampler* is illustrated in Algorithm 2 of Appendix C.

### 4.2 ADAPTIVE NEGATIVE SAMPLING

The *RejectSampler* can help to alleviate the class-imbalance issue. We next introduce the full VINS approach, which considers the **dynamic relative rank position of positive and negative** items for finding more informative negative samples avoiding $\lambda_{ij}^u \to 0$ as much as possible, which is very important for dealing with the mentioned gradient vanishment issue in Section 3.2. Algorithm 3 in Appendix C presents VINS in details. Specifically, to generate a negative sample, *RejectSampler* is firstly used to sample an item that is not connected to user $u$ (line 5 to 7 in the algorithm). Note that item $j$ sampled from *RejectSampler* is not guaranteed to be negative for user $i$. Therefore *RejectSampler* is re-called if $j$ is connected to $u$ ($e_{uj} \in E$). The next step is to evaluate if the sampled item $j$ is a violated one, which satisfies $\epsilon + x_{uj} \geq x_{ui}$, where $\epsilon$ is a margin (line 8 to 13 in the algorithm). In fact, there can be a set of violated negative samples, noted as $\mathcal{V}_i^u = \{j | \epsilon + x_{uj} \geq x_{ui}, e_{uj} \notin E\}$. The hardness of searching a violated negative sample increases when the positive item $i$ is ranked higher. This hardness is reflected as the weight factor $w_{ui}$ in Equation (1). For the positive item $i$ with a relative high rank position, we should generate a small weight $w_{ui}$, while give large weights to those lower-ranked ones. We thus define the weight as $w_{ui}(r_i)$, where $r_i = \sum_{j \in V_i^u} \pi(j) \mathbb{I}(\epsilon + x_{uj} \geq x_{ui})$ is the rank-aware variable of item $i$. $\mathbb{I}(x)$ is an indicator function. From the definition of $r_i$, we can see that the smaller $r_i$ is, the high-rank position of item $i$ is. Previous work (Weston et al., 2011) takes a truncated Harmonic Series function to generate the weight $w_{ui}(r_i) = \sum_{s=1}^{r_i} \frac{1}{s}$. We can see that Harmonic Series weighting method needs calculate the summation term for each given estimated rank position, which has the worst complexity $O(|I|)$ for each sample. Inspired by the lower bound of truncated Harmonic Series (shown in Lemma 4.1), we derive a efficient way to calculate the $w_{ui}(r_i)$ with complexity $O(1)$.

**Lemma 4.1.** *For a given $k \in \mathbb{N}$, truncated Harmonic Series $\sum_{s=1}^{2^k} \frac{1}{s} \geq 1 + \frac{k}{2}$. When $k = 0$, the equality holds.*

The complete proof of Lemma 4.1 is described in the Appendix D. According to this lemma, we divide the rank list into $k$ chunks, each of which has size $2^{k-1}$ (growing with the chunk number $k$), and is attached with a weight (*i.e.* 0.5). For the rank variable $r_i$, we can derive the chunk number based on the Geometric progression formulation, which leads to $\lceil * \rceil log_2(r_i + 1)$. Since first chunk has weight 1, we subtract 1 in the definition. More specifically, we define $w_{ui}(r_i)$ as follows: $w_{ui}(r_i) = \frac{1+0.5 \cdot (\lceil * \rceil \log_2(r_i+1)-1)}{1+0.5 \cdot (\lceil * \rceil \log_2(Z+1)-1)}$, where 0.5 is the weight for each chunk with size $2^{k-1}$, and $Z = \sum_{i \in I} \pi(i)$. However, $r_i = \sum_{j \in V_i^u} \pi(j)$ is difficult to attain. We use an item buffer $buffer_{ui}$ with size $\kappa$ to store every sampled negative candidate $j$. Then, $r_i$ can be approximated as $r_i \approx \lfloor \frac{Z}{min(K,\kappa)} \rfloor$, where $K$ is the number of steps to find item $j$. The final informative negative item $j$ to update model parameters will be selected from the top of the sorted $buffer_{ui}$ in descending order based on $x_{uj}$, as shown in Algorithm 1 of Appendix C. With the selected negative item $j$ by VINS, we can construct pairwise sample $(u, i, j)$ to train the ranking model. The employment of *RejectSampler* in VINS has two benefits. First, it considers the class-imbalance issue and tends to select the useful negative items than doing randomly, given the fact that items with large imbalance values usually have large norm that makes them difficult to be distinguished from positive items. Second, it reduces the size of negative item candidate set to explore through selecting the useful negative samples to the $buffer$. More discussion about the characteristics of the proposed method and its connection to previous methods can be found in Appendix E.

## 5 EXPERIMENTS

In this section, we conduct extensive experiments to answer three research questions: [**RQ1**] How will the item imbalance value evolve when using different sampling strategies? [**RQ2**] What are the advantages of VINS, comparing with the state-of-the-art baselines? [**RQ3**] How VINS can improve the computationally expensive models by sampling the most useful training data?

### 5.1 EXPERIMENTAL SETTINGS

**Datasets and Evaluation Metrics.** To validate the proposed sampling method, we use four publicly available datasets, from Yelp Challenge (13th round) [1], Amazon [2] and Steam (Kang and McAuley, 2018). The detailed information about the datasets and the way to obtain the training/testing dataset are reported in Appendix F1. We evaluate all of algorithms by top-$N$ ranking metrics including **F1** (Karypis, 2001), **NDCG** (Weimer et al., 2008).

**Recommenders.** In this work, we mainly study the state-of-the-art sampling methods in terms of their effectiveness and efficiency. To uncover the features of different samplers, we consider representative factorization models (MF (Rendle et al., 2009) and FPMC (Rendle et al., 2010)) and one state-of-the-art deep model (MARank (Yu et al., 2019)) which can capture users' temporal dynamic preferences. The details of these recommendation models are described in Appendix F2.

**Baselines/Negative Samplers & Pairwise Loss.** The baselines include **Uni** (Rendle et al., 2009) sampling a negative item from uniform distribution, **POP** (Mikolov et al., 2013) sampling negative items from a given distribution $\pi$, relative-order sampling methods, **Dynamic Negative Sampling (DNS)** (Zhang et al., 2013), **LFM-D** (Yuan et al., 2016) and **LFM-W** (Yuan et al., 2016), **AOBPR** (Rendle and Freudenthaler, 2014), **CML** (Hsieh et al., 2017), adversarial-like methods (**SA**) (Sun et al., 2019), **PRIS** (Lian et al., 2020), and **IRGAN** (Wang et al., 2017). Since the samplers are independent of the specific recommenders to work with, **we take MF as the base model to study their features, then switch to more complicated models (*i.e.*, FPMC, MARank)**. To keep the consistency of experimental setting for different baselines except IRGAN, we instantiate ranking objective $\ell(\cdot)$ as pairwise ranking loss (Rendle et al., 2009) for all baselines used in this work. The implementation detail of each method can be found in Appendix F3.

### 5.2 ITEM IMBALANCE VALUE EVALUATION (RQ1)

To evaluate the *item imbalance value* when applying different sampling methods, we count the number of appearance in positive and negative samples for each item. Then we track the evolution of

---

[1]https://www.yelp.com/dataset/challenge
[2]http://jmcauley.ucsd.edu/data/amazon/

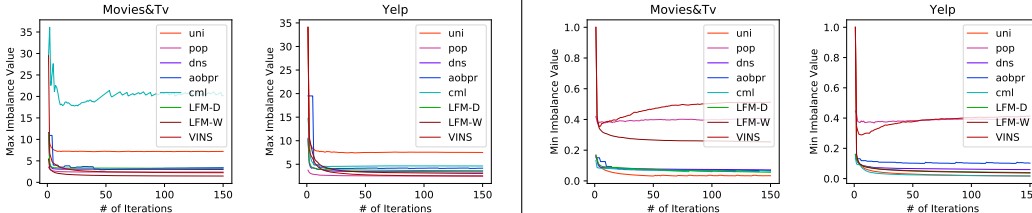

Figure 5: Evolution of maximum/minimum imbalance value of different sampling methods.

Table 1: Ranking performance when using different sampling methods with MF as the recommender for top-10 recommendation. The best is marked with underline, the second best is marked by *.

| Method | Sampler | Yelp | | Movies&Tv | | CDs&Vinyl | | Steam | |
|---|---|---|---|---|---|---|---|---|---|
| | | F1@10 | NDCG@10 | F1@10 | NDCG@10 | F1@10 | NDCG@10 | F1@10 | NDCG@10 |
| Item-KNN | | 0.0153 | 0.0205 | 0.0178 | 0.0258 | 0.0191 | 0.0261 | 0.0296 | 0.0409 |
| MF | Uni | 0.0135 | 0.0168 | 0.0146 | 0.0186 | 0.0195 | 0.0249 | 0.0338 | 0.0457 |
| | POP | 0.0129 | 0.0161 | 0.0179 | 0.0232 | 0.0229 | 0.0301 | 0.0333 | 0.0472 |
| | AOBPR | 0.0140 | 0.0173 | 0.0153 | 0.0197 | 0.0211 | 0.0278 | 0.0334 | 0.0463 |
| | CML | 0.0177 | 0.0216 | 0.0133 | 0.0179 | 0.0205 | 0.0276 | 0.0239 | 0.0317 |
| | PRIS | 0.0158 | 0.0210 | 0.0184 | 0.0239 | 0.0252 | 0.0331 | 0.0374 | 0.0502 |
| | SA | 0.0161 | 0.0199 | 0.0159 | 0.0206 | 0.0243 | 0.0326 | 0.0347 | 0.0483 |
| | IRGAN | 0.0188 | 0.0235 | 0.0206 | 0.0269 | 0.0263 | 0.0348 | 0.0358 | 0.0512 |
| | DNS | *0.0197 | *0.0247 | *0.0211 | *0.0276 | *0.0275 | *0.0366 | 0.0398 | 0.0551 |
| | LFM-D | 0.0187 | 0.0234 | 0.0204 | 0.0267 | 0.0269 | 0.0354 | *0.0406 | *0.0561 |
| | LFM-W | 0.0202 | 0.0255 | 0.0236 | 0.0313 | 0.0301 | 0.0401 | 0.0414 | 0.0569 |
| | VINS (ours) | **0.0222** | **0.0281** | **0.0245** | **0.0326** | **0.0310** | **0.0410** | **0.0429** | **0.0594** |
| Improvement | ours vs best | 9.9% | 10.2% | 3.81% | 4.15% | 2.99% | 2.24% | 3.62% | 4.39% |
| | ours vs second | 12.7% | 13.7% | 16.1% | 18.1% | 12.7% | 12.0% | 5.66% | 5.88% |

the maximum and minimum imbalance value. Due to the characteristics of adversarial-like methods themselves such as SA, PRIS, IRGAN, it's difficult to catch the evolution of items' imbalance value. Therefore, we discard them and focus on the other methods. It is expected that non-uniform sampling methods can downgrade the maximum but increase the minimum imbalance value comparing with the UNI method. From the results shown in Figure 5 (results on other two datsets can be found in Figure 9 of Appendix F4), we can find that most of baselines reach the expectation. Combining with the overall performance shown in Table 1, we can see that all of the methods outperform the UNI method. From this point of view, alleviating the class-imbalance issue has positive effect on the performance of learned model. It's also consistent with theoretical analysis in previous sections. It's easy to understand that POP method reaches the expectation, because we already have empirical analysis result in Figure 2. However, dynamic sampling methods like DNS, LFM-W did not have a clear statement on sampling from a non-uniform distribution. If we connect this finding with the logic relation between *class-imbalance* issue and *imbalanced item theorem*, it could provide an important clue to understand that bias to items with larger prediction value will tend to sample imbalanced items. The proposed method VINS does not ideally increase the minimum class-imbalance value in Steam data. However, VINS keeps imbalance value larger than the other methods except POP, and with the help of adaptive sampling strategy, VINS achieves better performance than the baselines from the results shown in Table 1. From this point of view, alleviating the class-imbalance issue has positive effect on the performance of learned model. It's also consistent with our theoretical analysis in previous sections.

## 5.3 RANKING PERFORMANCE (RQ2)

Table 1 summarizes the ranking performance of different sampling methods when applied to optimizing the same objective function. Dynamic sampling methods LFM-W and VINS significantly outperform the other baselines with a clear margin. While, the proposed sampler VINS is superior to the state-of-the-art method LFM-W. In terms of negative candidate selection, LFM-W depends on uniform distribution without any knowledge about the item property, while VINS selects the negative candidates with reject probability depending on item degree distribution. Ideally VINS can be more easier to obtain a violated negative sample than LFM-W according to Theorem 3.1 and 3.2. The experimental results validate the effectiveness of VINS which selects the negative candidates with reject probability motivated by class-imbalance issue.

## 5.4 TIME COMPLEXITY ANALYSIS (RQ2)

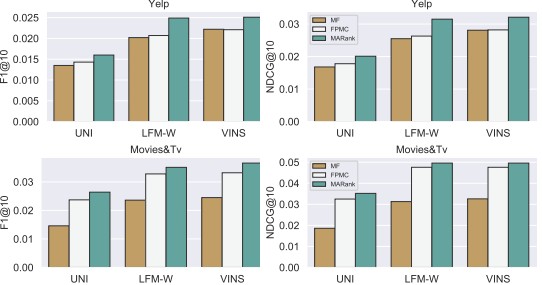

Figure 6: Time complexity with the growth of model complexity.

From Table 2, we can see that as the data scale up in size, all samplers will need more time. Especially, LFM-W needs over 10x more time comparing with stationary sampling methods, while VINS is more efficient than LFM-W. The average number of steps to find a violated negative sample is the key for the time complexity analysis. As discussed in Section 1, time complexity of dynamic sampling approaches like LFM-W and VINS heavily depends on the search of a proper negative sample from massive trials. To further investigate the sampling process, we use the buffer technology for both methods to show the connection between the model performance convergence

Table 2: Time complexity comparison with different data scale in terms of average running time per epoch in minutes and ratio to the simplest method "Uni".

| Sampler | Steam (smallest) | CDs&Vinyl | Movies&Tv | Yelp (largest) |
|---|---|---|---|---|
| Uni | 0.07 (1x) | 0.1 (1x) | 0.16 (1x) | 0.47 (1x) |
| POP | 0.09 (1.28x) | 0.13 (1.3x) | 0.2 (1.25x) | 0.58 (1.23x) |
| AOBPR | 0.05 (0.71x) | 0.23 (2.3x) | 0.32 (2x) | 1.98 (4.21x) |
| CML | 0.33 (4.71x) | 0.33 (3.3x) | 0.5 (3.1x) | 1.23 (2.61x) |
| PRIS | 1.12 (16x) | 1.38 (13.8x) | 2.07 (12.9x) | 6.25 (13.3x) |
| SA | 0.48 (6.85x) | 0.66 (6.6x) | 0.92 (5.75x) | 2.76 (5.87x) |
| IRGAN | 3.85 (55x) | 4.54 (45.4x) | 5.6 (35x) | 23.4 (49.8x) |
| DNS | 0.28 (4x) | 0.44 (4.4x) | 0.72 (4.5x) | 2.1 (4.46x) |
| LFM-D | 0.38 (5.42x) | 0.49 (4.9x) | 1.1 (6.87x) | 1.86 (3.95x) |
| LFM-W | 0.35 (5x) | 1.58 (15.8x) | 1.65 (10.3x) | 4.78 (10.1x) |
| VINS | 0.25 (3.57x) | 1.08 (10.8x) | 1.12 (6.37x) | 3.05 (6.48x) |

and maximum steps to sample a violated item. The original LFM-W did not define a buffer, we set the maximum of sampling trials for **LFM-W** to 1024. The results shown in Table 5 of Appendix F4 demonstrate that VINS can converge to stable performance with less trials for each positive sample, while LFM-W needs a larger buffer with at least 1024 slots. From the results shown in Table 4 of Appendix F4, we can see that VINS can converge to the better solution than LFM-W, meanwhile needs only a small number of trials to find a violated item. This leads to over 30% training time saved comparing to LFM-W, shown in Table 2.

## 5.5 Performance on Computationally Expensive Methods (RQ3)

By far, we only apply the dynamic sampling methods on a linear recommendation model (MF). It is also interesting to evaluate their performance on more complicated models, for example FPMC and MARank, for next-item prediction. From the experimental results shown in Figure 6 we can find that VINS can save more training time (from 50% to 60%) than LFM-W ranging from shallow model FPMC to deep attentive model MARank, while reaching the best recommendation performance shown in Figure 7 and 10 (Appendix F4). This significant acceleration of recommendation model

Figure 7: Ranking performance on F1/NDCG metric of shallow and deep models.

training verifies that VINS is an effective dynamic negative sampling method. Especially for deep neural models training, VINS is a promising tool to select the most useful negative samples for achieving both significant reduction of training time and improvement of inference capability.

## 6 Conclusions

In this work, we systematically study the class-imbalance problem in pairwise ranking optimization for recommendation tasks. We indicate out the edge- and vertex-level imbalance problem, and show its connection to sampling a negative item from static distribution. To tackle the challenges raised by the class-imbalance problem, we propose a two-phase sampling approach to alleviate the imbalance issue by tending to sample a negative item with a larger degree and close prediction score to the given positive sample. We conduct thorough experiments to show that the biased sampling method with reject probability can help to find violated samples more efficiently, meanwhile having a competitive or even better performance with state-of-the-art methods.

# 7 REPRODUCIBILITY STATEMENT

The detailed information about the reproducibility is presented in Appendix F3. The way to obtain the experimental data can be found in Appendix F1. The pseudo-code of the proposed algorithm can be found in Appendix C.

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

APPENDIX

A. PROOF FOR THEOREM 3.1

*Proof.* With the given user-item graph $G$, both $\sum_{j \in I} \pi(j)$ and $|E|$ are constant. Let's define a function $f(x) = \frac{x^{1-\beta} \cdot c_1}{c_2 - x}$, where $c_1 = \sum_{j \in I} \pi(j)$ and $c_2 = |E|$. Then we can have first-order derivative $\nabla f(x) > 0$, which means $IV(i) > IV(j)$ if $d_i > d_j$. $\qquad\square$

B. PROOF OF THEOREM 3.2

*Proof.* Given latent space with $d$ dimensions, there exists $d$ - 1 mutually orthogonal vectors $\vec{c}_2, \vec{c}_3, \cdots, \vec{c}_d$ and $\vec{c}_1 = \hat{P}^\tau$. Let $\Delta_i^+(t) = \sum_{u \in \mathcal{N}_i} \frac{\partial \ell_{ij}^u}{\partial x_{ui}} P_u^t$ denote the gradients received when item $i$ acted as a positive sample, and $\Delta_i^-(t) = -\sum_{u \in \mathcal{N}_i^-} \frac{\partial \ell_{ij}^u}{\partial x_{ui}} P_u^t$ denote the gradients received when acting as a negative sample. It's noted that if item $i$ has a large imbalance value, the size of $|\mathcal{N}_i|$ is usually $\gg |\mathcal{N}_i^-|$, and vice versa. Then for any iteration $n > \tau$, the embedding of item $i$ is updated with gradient descent method as:

$$P_i^n = P_i^\tau + \eta \sum_{n \geq t > \tau} (\Delta_i^+(t) + \Delta_i^-(t))$$

Then we can perform coordinate axis transform on $P_i^t$ and $P_u^t$ to $c_1, \cdots, c_d$.

$$\Rightarrow P_i^n = \alpha_i^1 \vec{c}_1 + \cdots + \alpha_i^d \vec{c}_d$$
$$+ \eta \sum_{n \geq t > \tau} \sum_{u \in \mathcal{N}_i} \frac{\partial \ell_{ij}^u}{\partial x_{ui}}(t)(\beta_u^1 \vec{c}_1 + \cdots + \beta_u^d \vec{c}_d)$$
$$- \eta \sum_{n \geq t > \tau} \sum_{u \in \mathcal{N}_i^-} \frac{\partial \ell_{ij}^u}{\partial x_{ui}}(t)(\gamma_u^1 \vec{c}_1 + \cdots + \gamma_u^d \vec{c}_d)$$

Now we have $P_i^\tau = \alpha_i^1 \vec{c}_1 + \cdots + \alpha_i^d \vec{c}_d$ and $P_u^t = \beta_u^1 \vec{c}_1 + \cdots + \beta_u^d \vec{c}_d$, $\frac{\partial \ell_{ij}^u}{\partial x_{ui}}(t) > 0$, $\beta_u^1 > 0$ as inner-product $< P_u^t, \vec{c}_1 > = < P_u^t, \hat{P}_u^\tau >$, and all other variables $\in \mathbb{R}$.

$$\Rightarrow P_i^n = \alpha_i^1 \vec{c}_1 + \cdots + \alpha_i^d \vec{c}_d + \sum_{n \geq t > \tau} \lambda_1(t) \vec{c}_1 + \cdots + \lambda_d \vec{c}_d,$$

where $\lambda_k(t) = \eta\left(\sum_{u \in \mathcal{N}_i} \frac{\partial \ell_{ij}^u}{\partial x_{ui}}(t)\beta_u^k - \sum_{u \in \mathcal{N}_i^-} \frac{\partial \ell_{ij}^u}{\partial x_{ui}}(t)\gamma_u^k\right)$ for $k \in [1, d]$. Since coordinates $\vec{c}_1, \vec{c}_2, \cdots, \vec{c}_d$ are manually orthogonal.

$$\Rightarrow \lim_{n \to \infty} ||P_i^n||^2 = \lim_{t \to \infty} (\alpha_i^1 + \sum_{n \geq t > \tau} \lambda_1(t))^2 ||\vec{c}_1||^2 + \cdots$$
$$+ (\alpha_i^d + \sum_{n \geq t > \tau} \lambda_d(t))^2 ||\vec{c}_d||^2$$
$$\geq \lim_{n \to \infty} (\alpha_i^1 + \sum_{n \geq t > \tau} \lambda_1(t))^2 ||\vec{c}_1||^2$$
$$\geq \lim_{n \to \infty} (\alpha_i^1 + (n - \tau) \cdot min_{n \geq t > \tau} \lambda_1(t))^2 ||\vec{c}_1||^2$$

And we have

$$\lambda_1(t) = \eta\left(\sum_{u \in \mathcal{N}_i} \frac{\partial \ell_{ij}^u}{\partial x_{ui}}(t)\beta_u^1 - \sum_{u \in \mathcal{N}_i^-} \frac{\partial \ell_{ij}^u}{\partial x_{ui}}(t)\gamma_u^1\right),$$
$$where \quad \frac{\partial \ell_{ij}^u}{\partial x_{ui}}(t)\beta_u^1 > 0$$

For imbalanced items, the value of $\lambda_1(t)$ will be dominated by the size of $\mathcal{N}_i$ and $\mathcal{N}_i^-$. If an imbalanced item with a very large imbalance value, then we could have $\lambda_1(t) > 0$ with a relative high probability. Then we have $\lim_{n \to \infty} (\alpha_i^1 + (n - \tau) \cdot min_{n \geq t > \tau} \lambda_1(t))^2 ||\vec{c}||^2 = \infty$. $\qquad\square$

## C. PSEUDOCODE

The detailed implementation of the proposed method VINS and its key component *RejectSampler* can be found in Algorithm 1 and 2, respectively.

---

**Algorithm 1:** VINS

---

1  **Input:** $G = (V, E)$, max step $\kappa$, positive pair $(u, i)$, max shot s, margin $\epsilon$
2  **Output:** negative item $j$, and $w_{ui}(r_i)$
3  $selected_j = -1$, $max_j = -inf$
4  **for** $K \leftarrow 1$ **to** $\kappa$ **do**
5      **do**
6          |   $j = RejectSampler(i, s, \pi)$
7      **while** $e_{uj} \in E$;
8      $x_{uji} = x_{uj} + \epsilon - x_{ui}$
9      **if** $x_{uj} > max_j$ **then**
10         $max_j = x_{uj}$
11         $selected_j = j$
12     **if** $x_{uji} > 0$ **then**
13         break;
14 $r_i = \lfloor \frac{Z}{min(K, \kappa)} \rfloor$;
15 **return** $selected_j, w_{ui}(r_i)$;

---

---

**Algorithm 2:** REJECTSAMPLER

---

1  **Input:** item $i$, max shot $s$, weight distribution $\pi$
2  **Output:** selected item $j$
3  $selected_j = $ -1, $maxi_j = $ -1
4  **for** $iter \leftarrow 1$ **to** $s$ **do**
5      j = randint(Z);
6      // in case of the extreme popular item i
7      **if** $\pi(j) > maxi\_deg$ **then**
8          $maxi_j = \pi(j)$;
9          $selected_j = $ j;
10     reject_ratio = 1 - min $\{\frac{\pi(j)}{\pi(i)}, 1\}$;
11     **if** *random.uniform() > reject_ratio* **then**
12         $selected_j = $ j;
13         break;
14 **return** $selected_j$;

---

## D. PROOF FOR LEMMA 4.1

*Proof.*

$$\sum_{s=1}^{2^k} \frac{1}{s} = 1 + \frac{1}{2} + \frac{1}{3} + \frac{1}{4} + \frac{1}{5} + \frac{1}{6} + \frac{1}{7} + \frac{1}{8} + \cdots + \frac{1}{2^k}$$

$$\geq 1 + \underbrace{\frac{1}{2}}_{1} + \underbrace{(\frac{1}{4} + \frac{1}{4})}_{2^1} + \underbrace{(4 \times \frac{1}{8})}_{2^2} \cdots \underbrace{(\frac{2^{k-1}}{2 \times 2^{k-1}})}_{2^{k-1}}$$

$$\geq 1 + \frac{k}{2}$$

$\square$

E. DISCUSSION ON VINS

E1. COMPLEXITY DISCUSSION

The most computationally expensive part of the proposed VINS model is the relative-order sampling procedure (line 4 to 13 in Algorithm 3). As discussed previously, finding a violated sample needs iterative comparison of the prediction value between a positive item and a negative item candidate. For each negative sample, the computation complexity is $O(d)$, where $d$ is the embedding size. Assume that the average number of steps to obtain a violated negative item is $h'$ and the maximum number of chances to reject a sampled item from the *RejectSampler* is $s$, then the time complexity of VINS will be $O(|E| \cdot (d+s) \cdot h')$. Usually, $s \ll d$ can be a very small number. Therefore, comparing the proposed approach with the state-of-the-art dynamic sampling method (Yuan et al., 2016), the time complexity difference will be the average number of steps $h'$ to find a violated item. From the experimental analysis, we find that the proposed *RejectSampler* significantly speeds up searching a violated sample.

E2. CONNECTION TO EXISTING APPROACHES

Most of negative sampling approaches assume that the negative items follow a pre-defined distribution $Q(j)$. According to the strategies to obtain a negative item, we can summarize the main kinds of negative samplers into three categories: user-independent, user-dependent, edge-dependent. The proposed approach (VINS) can be regarded as a general version of several methods by controlling the setting of hyper-parameters $\{\kappa, \beta\}$.

- *user-independent*: As the representatives, UNI (Rendle et al., 2009) and POP (Mikolov et al., 2013) initialize the $Q(j)$ as a static distribution $\boldsymbol{\pi}$. VINS can actually implement these two methods by setting $\kappa = 1, \beta = 0$ for UNI, and $\kappa = 1, \beta \in [0, 1]$ for POP.

- *user-dependent*: This type of methods usually define a conditional distribution $Q(j|u)$ which can capture the dynamics of learning procedure to some extent. Sampling from the exact distribution $Q(j|u)$ will cost massive number of time in large-scale item database. Most of methods turn to defining a sub-optimal distribution based on a small number of candidate set. For example, DNS (Zhang et al., 2013) greedily selects the item with the largest predicted score $x_{uj}$ from the candidate set. Self-adversarial (SA) (Sun et al., 2019) method first sample candidates from uniform distribution, then calculate the weight of candidate through a $softmax(x_{uj})$ distribution. Similar idea can be found in more recent proposed method PRIS (Lian et al., 2020). While, PRIS tries to select a negative sample from the distribution $Q(j|u)$ through a importance sampling approach. By borrowing ideas from GAN, IRGAN (Wang et al., 2017) propose a two-agent minmax games, where generator $G$ aborbs knowledge from discriminator, then selects negative samples from $Q_G(j|u) = softmax(x_{uj})$ to update discriminator. From the view of distribution alignment, the generator actually attempts to learn distribution from the discriminator by taking *Reinforcement Learning* (RL) as the workhorse. However, RL methods usually need lots of training cases to update their policy, and sampling according to the policy distribution relies on the exact distribution $Q_G(j|u)$ over the whole item set, which makes IRGAN become very slow to converge and difficult to tune the model. Moreover, the generator might have a distribution which could delay from the discriminator, which can lead to unqualified negative samples produced by the generator.

- *edge-dependent*: The methods mentioned above do not consider a fact that the ranking position of positive item *i* evolves as the learning procedure move forwards, in other words, the informative negative item set also changes. The edge-dependent methods aim at selecting informative negatives from distribution $Q(j|u, i)$. As an initial study, Weston *et al.* (Weston et al., 2011) proposed the WARP loss by designing a rank-aware distribution $r_i = \sum_{j \in V_i^u} \mathbb{I}(\epsilon + x_{uj} \geq x_{ui})$. However, it's impossible to get the exact $r_i$ for every single training sample $(u,i)$ during the training stage. Fortunately the negative item *j* can be obtained through estimating a geometric distribution $P(X = k)$ parameterized with $p = \frac{r_i}{Z}$. There're many works that are based on WARP and all of them follow the same idea as WARP to estimate the $P(X = k)$ from a uniform distribution. VINS also inherits the basic ideas from WARP but modifies the target distribution as $r_i = \sum_{j \in V_i^u} \pi(j) \mathbb{I}(\epsilon + x_{uj} \geq x_{ui})$, and proposes to estimate it through an importance sampling method after theoretically investigating the existing class-imbalance issue and its potential influence. As the state-of-the-art variant of WARP loss, LFM-W advances WARP with a normalization

term. However, estimating the geometric distribution from a uniform distribution makes LFM-W need lots of steps to find a violated sample. Moreover, LFM-W might find sub-optimal negative sample without considering the class-imbalance issue. LFM-W can be equivalent to VINS by setting $\beta = 0$ and replacing the weight function $w_{ui}(r_i)$ as a truncated Harmonic Series function, *i.e.* $w_{ui}(r_i) = \sum_{z=1}^{\lceil r_i \rceil} \frac{1}{z}$.

## E3. Rank Estimation Bias Discussion

Let $Pr(X = k) = p(1 - p)^{k-1}$ denote geometric distribution with parameter $p$, and $\{X_1, X_2, \cdots, X_n | X_i \in \mathbb{N}\}$ to be the observations. The optimized estimation of $p$ by maximizing the likelihood function will be $\hat{p} = 1/\overline{X}$, where $\overline{X} = \sum_{i=1}^{n} X_i/n$. Now we can obtain the expectation over the estimated parameter, *i.e.* $E[\hat{p}] = E[1/\overline{X}]$. When estimating the rank-aware variable $r_i$, we usually conduct single experiment, that is $n = 1$. In Lemma 7.1, we demonstrate that it lead to a biased estimation of the true rank position. Fortunately, we find that the estimation error will become smaller as the positive sample get better and better ranking position as the learning procedure move forwards.

**Lemma 7.1.** *For special case $n = 1$, $E[\hat{p}]$ will be larger than* p, *which means $\hat{p}$ is not an unbiased estimation.*

*Proof.*

$$
\begin{aligned}
E[\hat{p}] = E[1/X_1] &= \sum_{k=1}^{\infty} \frac{1}{k} p(1-p)^{k-1} \\
&= p + \sum_{k=2}^{\infty} \frac{1}{k} p(1-p)^{k-1}
\end{aligned}
\tag{2}
$$

for $p \in (0, 1)$ in this case, the above sum term is strictly positive. □

Since we can get the estimated rank position as $\hat{r}_i = \hat{p}_i \cdot Z$. To save computational cost, we usually run one time *i.e.*, $n = 1$ to estimate the mass variable in dynamic sampling approach. Under this scheme, the estimation expectation $E[\hat{r}_i] = E[\hat{p} \cdot Z] = E[Z/X_1]$. If we fold out this equation, we can get the following induction:

$$
\begin{aligned}
E[\frac{Z}{X_1}] &= Z \sum_{k=1}^{\infty} \frac{1}{k} p(1-p)^{k-1} \\
&= r_i + \sum_{k=2}^{\infty} \frac{r_i}{k} (1 - \frac{r_i}{Z})^{k-1} > r_i
\end{aligned}
\tag{3}
$$

where $r_i = Z \cdot p$ denotes ground truth value. Let $h(r_i) = r_i + \sum_{k=2}^{\infty} \frac{r_i}{k} (1 - \frac{r_i}{Z})^{k-1}$ represent a function of $r_i$. It's very hard to analyze the gradients of function $h(\cdot)$. However, we need answer what's the exact estimation bias as the change of idea ranking $r_i$. To answer this question, we turn to analyze a ratio function $\psi(r_i) = (h(r_i) - r_i)/r_i = \sum_{k=2}^{\infty} \frac{1}{k} (1 - \frac{r_i}{Z})^{k-1}$. Comparing to directly analyzing original function $h(r_i)$, $\psi(r_i)$ is a monotone decreasing function. Based on the feature, we empirically illustrate the change of estimation bias ratio and the rank variable $r_i$. From Figure 8 we can see that as the item ranks higher, the estimation error will be smaller.

## E4. False negative issue

With the given partial knowledge, in particular only implicit feedback from users, it's very difficult to discriminate false negative and true negative items. It has become a common challenge for designing a negative sampler (Ding et al., 2020). In this paper, we do not focus on studying how to overcome the "false negative issue", but the proposed adaptive strategy could have a mechanism to avoid to push "false negative" item far away from the users. According to definition of weight $w_{ui}(r_i)$, the more difficult to sample a violated negative item, the smaller weight $(w_{ui}(r_i) \to 0)$ it is, which suggests that the rank position of positive item $i$ is learned well. For this item $i$, even the sampled item is a false negative, it will get a very small gradient (almost zero gradient) to move away from the target user. We'd like to leave this challenge for future work.

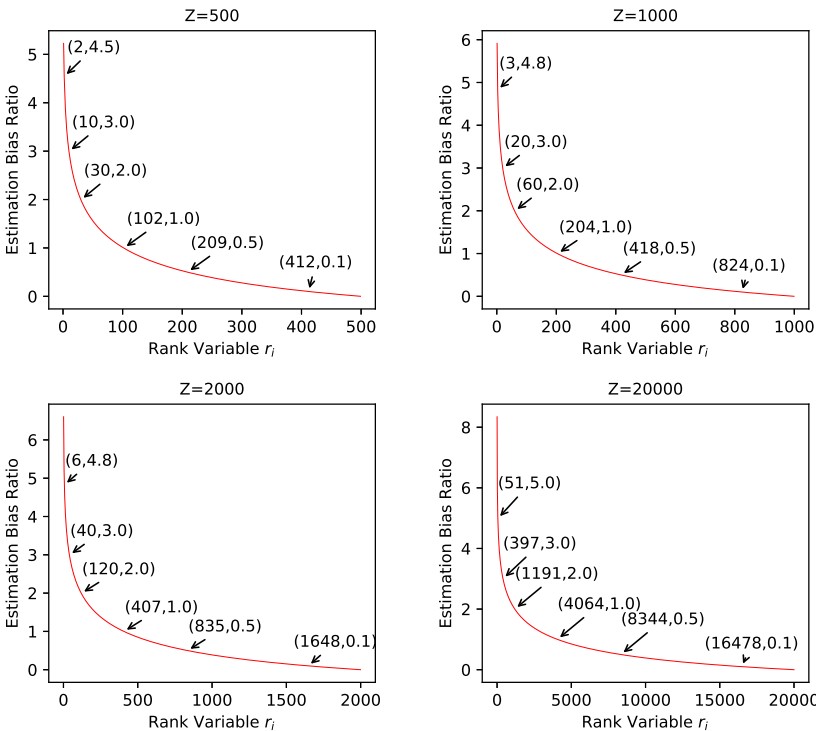

Figure 8: The evolution of estimated rank variable $\psi(r_i)$.

Table 3: Statistical information of the datasets.

| Data | #Users | #Items | #Observation | Sparsity |
|---|---|---|---|---|
| Yelp | 113,917 | 93,850 | 3,181,432 | 99.97% |
| Movies&Tv | 40,928 | 51,509 | 1,163,413 | 99.94% |
| CDs&Vinyl | 26,876 | 66,820 | 770,188 | 99.95% |
| Steam | 20,074 | 12,438 | 648,202 | 99.74% |

## F. EXPERIMENTAL SETTING AND ANALYSIS

### F1. DATASETS

The detailed statistics of the employed datasets can be found in Table 3. Following the processing in (Tang and Wang, 2018; He and McAuley, 2016), we discard inactive users and items with fewer than 10 feedbacks since cold-start recommendation usually is regarded as a separate issue in the literature (He and McAuley, 2016; Rendle et al., 2010). For each dataset, we convert star-rating into binary feedback regardless of the specific rating values since we care more about the applications without explicit user feedbacks like ratings (He et al., 2017a; 2016). We split all datasets into training and testing set by holding out the last 20% review behaviors of each user into the testing set, the rest as the training data.

### F2. RECOMMENDERS

- **Matrix Factorization** (MF) (Rendle et al., 2009): This method uses a basic matrix factorization model as the scoring function. It can be regarded as a shallow neural network with a single hidden layer which takes user and item one-hot vector as input (He et al., 2017b).

- **Factorizing Personalized Markov Chains** (FPMC) (Rendle et al., 2010): It's a method that combines the MF and factorized Markov Chain over item sequence for next-item prediction.

- **MARank** (Yu et al., 2019): It incorporates both individual- and union-level item relation into a deep multi-order attentive encoder, instead of only using factorized item transition probability.

## F3. REPRODUCIBILITY

All methods are optimized with Adam and implemented in Tensorflow with a GeForce GTX 1080Ti GPU. We share the parameter setting of the optimizer for all baselines and experiments in this work, with default learning rate $\eta = 0.001$. We use grid search to examine the hyper-parameters, including the embedding size from $\{16, 64, 128\}$, $\lambda$ from $\{0.0005, 0.001, 0.005, 0.01\}$. Different baselines have their own hyper-parameters. For decay factor $\beta$ in POP sampler, the search space includes $\{0.25, 0.5, 0.75, 1\}$. Both CML and DNS need a number of negative candidates. In this work, a small number *e.g.*, 10 or 20 gives good enough results as suggested by the authors (Zhang et al., 2013; Hsieh et al., 2017). LFM-D needs two hyper-parameters, the number of negative candidates, and the expected sampling position. For the first one, it is the same as DNS, but usually needs a little larger number, *e.g.*, 20 in this work. The expected sampling position can be obtained by multiplying the number of negative candidates with a ratio factor $\rho$. The search space for $\rho$ was $\{0.01, 0.05, 0.1, 0.5\}$, and $\rho = 0.1$ gives the best results. AOBPR also needs to set the ratio factor $\rho$, and produces best results with $\rho = 0.1$. LFM-W only has a margin parameter $\epsilon$ besides the optimizer parameters and regularization term. This parameter actually varies as the type of employed optimizer and the validation model. We search the best choice $\epsilon$ from $\{1, 2, 3, 4\}$ for both LFM-W and VINS. For VINS, we need to search the best choice for buffer size $\kappa$ and decay factor $\beta$. In this work, we find that $\kappa = 64$ or 128 is good enough according to the analysis results. In terms of IRGAN, we implement this method with the published code [3] and suggested setting. In self-adversarial method (SA) [4], the discriminator and generator are the same prediction model. It creates an adversarial item by aggregating a number of negative items. In this work, we tried different settings from $\{64, 128, 256\}$, and select the best value *i.e.* 256. We follow the suggested setting by the authors to set up PRIS (Lian et al., 2020).

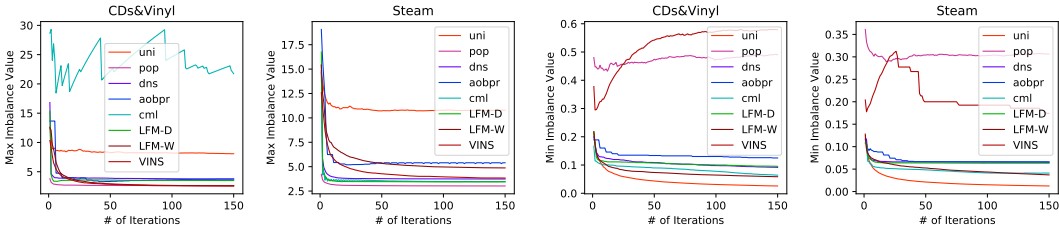

Figure 9: Evolution of maximum/minimum imbalance value of different sampling methods.

## F4. EXPERIMENTAL ANALYSIS

Due to the limited space in the main content, we will present additional experimental analysis results in this section. More specifically, we can find the changes of maximum/minimum imbalance value for the other two data in Figure 9. In Table 4 and 5, we can find the the comparison results between VINS and the best baseline LFM-W to show that the proposed method VINS is superior to LFM-W in terms of both efficiency and effectiveness. Figure 10 summarizes the additional ranking performance on the other two datasets.

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

Table 4: Time complexity analysis: number of average steps $h'$ to find a negative sample by LFM-W and VINS. The term behind $\pm$ stands for the standard variance.

| Epoch | 5 | 10 | 20 | 50 | 150 |
|---|---|---|---|---|---|
| | | | Yelp | | |
| LFM-W | 10.2±26.4 | 17.0± 52.2 | 19.8± 59.4 | 21.5± 63.2 | 21.7±65.2 |
| VINS | 8.7±14.5 | 11.8± 17.4 | 14.6± 19.7 | 16.2±21.0 | 16.3±21.0 |
| | | | Movies&Tv | | |
| LFM-W | 3.2±8.0 | 6.5±24.7 | 12.0±42.7 | 18.4±60.9 | 19.0±62.9 |
| VINS | 3.4±7.6 | 6.2±12.2 | 9.8±16.1 | 14.9±20.0 | 16.2±20.7 |
| | | | CDs&Vinyl | | |
| LFM-W | 3.5±15.1 | 10.0±40.6 | 17.1±58.2 | 28.5±83.3 | 29.3±85.5 |
| VINS | 3.8±10.1 | 7.9±14.8 | 12.8±18.8 | 21.4±23.2 | 23.4±23.9 |
| | | | Steam | | |
| LFM-W | 3.2±5.7 | 4.1±8.8 | 5.0±11.1 | 6.2±16.0 | 6.3±16.7 |
| VINS | 2.8±5.6 | 3.5±7.0 | 4.3±8.4 | 5.4±9.8 | 5.8±10.6 |

Table 5: Performances with different buffer size.

| Buffer Size | 8 | 16 | 32 | 64 | 128 | 1024 |
|---|---|---|---|---|---|---|
| | | | Yelp-F1@10 | | | |
| LFM-W | 0.0138 | 0.0164 | 0.0180 | 0.0189 | 0.0197 | 0.0202 |
| VINS | 0.0169 | 0.0185 | 0.0205 | 0.0222 | 0.0225 | 0.0223 |
| | | | Yelp-NDCG@10 | | | |
| LFM-W | 0.0185 | 0.0204 | 0.0224 | 0.0238 | 0.0251 | 0.0255 |
| VINS | 0.0209 | 0.0234 | 0.0253 | 0.0281 | 0.0284 | 0.0281 |
| | | | Movies&Tv-F1@10 | | | |
| LFM-W | 0.0193 | 0.0215 | 0.0223 | 0.0228 | 0.0232 | 0.0236 |
| VINS | 0.0222 | 0.0228 | 0.0235 | 0.0245 | 0.0243 | 0.0246 |
| | | | Movies&Tv-NDCG@10 | | | |
| LFM-W | 0.0252 | 0.0279 | 0.0295 | 0.0301 | 0.0305 | 0.0313 |
| VINS | 0.029 | 0.0302 | 0.0308 | 0.0326 | 0.0325 | 0.0326 |
| | | | CDs&Vinyl-F1@10 | | | |
| LFM-W | 0.0249 | 0.0270 | 0.0278 | 0.0296 | 0.0298 | 0.0301 |
| VINS | 0.0270 | 0.0285 | 0.0296 | 0.0310 | 0.0311 | 0.0312 |
| | | | CDs&Vinyl-NDCG@10 | | | |
| LFM-W | 0.0328 | 0.0352 | 0.0365 | 0.0392 | 0.0398 | 0.0401 |
| VINS | 0.0361 | 0.0376 | 0.0397 | 0.0402 | 0.041 | 0.0412 |
| | | | Steam-F1@10 | | | |
| LFM-W | 0.0389 | 0.0399 | 0.0404 | 0.0408 | 0.0409 | 0.0414 |
| VINS | 0.0408 | 0.0418 | 0.0426 | 0.0429 | 0.0430 | 0.0428 |
| | | | Steam-NDCG@10 | | | |
| LFM-W | 0.0533 | 0.0547 | 0.0552 | 0.0566 | 0.0568 | 0.0569 |
| VINS | 0.0567 | 0.0588 | 0.0603 | 0.0601 | 0.0601 | 0.0603 |

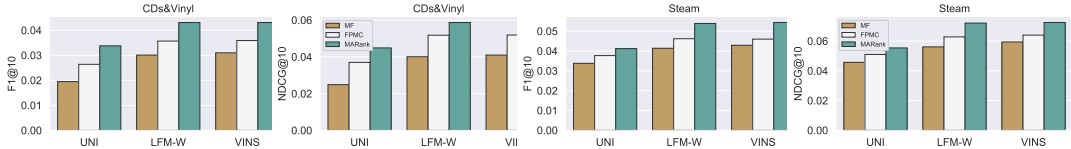

Figure 10: Ranking performance on F1/NDCG metric of shallow and deep models.

Jingtao Ding, Guanghui Yu, Xiangnan He, Fuli Feng, Yong Li, and Depeng Jin. 2019. Sampler design for bayesian personalized ranking by leveraging view data. *IEEE Transactions on Knowledge and Data Engineering (TKDE)* (2019).

Ruining He, Wang-Cheng Kang, and Julian McAuley. 2017a. Translation-based Recommendation. In *Proceedings of the eleventh ACM Conference on Recommender Systems (RecSys)*. 161–169.

Ruining He and Julian McAuley. 2016. Fusing similarity models with markov chains for sparse sequential recommendation. In *Proceedings of IEEE 16th International Conference on Data Mining (ICDM)*. 191–200.

Xiangnan He, Lizi Liao, Hanwang Zhang, Liqiang Nie, Xia Hu, and Tat-Seng Chua. 2017b. Neural collaborative filtering. In *Proceedings of the 26th international conference on world wide web*. 173–182.

Xiangnan He, Hanwang Zhang, Min-Yen Kan, and Tat-Seng Chua. 2016. Fast matrix factorization for online recommendation with implicit feedback. In *Proceedings of the 39th International Conference on Research and Development in Information Retrieval (SIGIR)*. 549–558.

Ko-Jen Hsiao, Alex Kulesza, and Alfred Hero. 2014. Social Collaborative Retrieval. In *Proceedings of the Seventh ACM International Conference on Web Search and Data Mining (WSDM)*. 293–302.

Cheng-Kang Hsieh, Longqi Yang, Yin Cui, Tsung-Yi Lin, Serge Belongie, and Deborah Estrin. 2017. Collaborative metric learning. In *Proceedings of the 26th International Conference on World Wide Web (WWW)*. 193–201.

Wang-Cheng Kang and Julian McAuley. 2018. Self-attentive sequential recommendation. In *Proceedings of the 2018 IEEE International Conference on Data Mining (ICDM)*. 197–206.

George Karypis. 2001. Evaluation of Item-Based Top-N Recommendation Algorithms. In *Proceedings of the 10th ACM on Conference on Information and Knowledge Management (CIKM)*. 247–254.

Guang-He Lee and Shou-De Lin. 2016. LambdaMF: Learning Nonsmooth Ranking Functions in Matrix Factorization Using Lambda. In *Proceedings of the 2016 IEEE International Conference on Data Mining (ICDM)*. 823–828.

Defu Lian, Qi Liu, and Enhong Chen. 2020. Personalized Ranking with Importance Sampling. In *Proceedings of The Web Conference 2020*. 1093–1103.

Tomas Mikolov, Ilya Sutskever, Kai Chen, Greg Corrado, and Jeffrey Dean. 2013. Distributed Representations of Words and Phrases and Their Compositionality. In *Proceedings of the Advances in Neural Information Processing Systems (NeurIPS)*. 3111–3119.

Steffen Rendle and Christoph Freudenthaler. 2014. Improving Pairwise Learning for Item Recommendation from Implicit Feedback. In *Proceedings of the Seventh ACM International Conference on Web Search and Data mining (WSDM)*. 273–282.

Steffen Rendle, Christoph Freudenthaler, Zeno Gantner, and Lars Schmidt-Thieme. 2009. BPR: Bayesian Personalized Ranking from Implicit Feedback. In *Proceedings of the Twenty-Fifth Conference on Uncertainty in Artificial Intelligence (UAI)*. 452–461.

Steffen Rendle, Christoph Freudenthaler, and Lars Schmidt-Thieme. 2010. Factorizing personalized markov chains for next-basket recommendation. In *Proceedings of the 19th international conference on World Wide Web (WWW)*. 811–820.

Zhiqing Sun, Zhi-Hong Deng, Jian-Yun Nie, and Jian Tang. 2019. RotatE: Knowledge Graph Embedding by Relational Rotation in Complex Space. In *Proceedings of the Seventh International Conference on Learning Representations (ICLR)*.

Jiaxi Tang and Ke Wang. 2018. Personalized Top-N Sequential Recommendation via Convolutional Sequence Embedding. In *Proceedings of the Eleventh ACM International Conference on Web Search and Data Dining (WSDM)*. 565–573.

Bo Wang, Minghui Qiu, Xisen Wang, Yaliang Li, Yu Gong, Xiaoyi Zeng, Jun Huang, Bo Zheng, Deng Cai, and Jingren Zhou. 2019. A Minimax Game for Instance Based Selective Transfer Learning. In *Proceedings of the 25th ACM SIGKDD International Conference on Knowledge Discovery and Data Mining (KDD)*. 34–43.

Jun Wang, Lantao Yu, Weinan Zhang, Yu Gong, Yinghui Xu, Benyou Wang, Peng Zhang, and Dell Zhang. 2017. Irgan: A minimax game for unifying generative and discriminative information retrieval models. In *Proceedings of the 40th International ACM SIGIR conference on Research and Development in Information Retrieval (SIGIR)*. 515–524.

Xiang Wang, Yaokun Xu, Xiangnan He, Yixin Cao, Meng Wang, and Tat-Seng Chua. 2020. Reinforced negative sampling over knowledge graph for recommendation. In *Proceedings of The Web Conference (WWW)*. 99–109.

Markus Weimer, Alexandros Karatzoglou, Quoc Viet Le, and Alex Smola. 2008. COFIRANK Maximum Margin Matrix Factorization for Collaborative Ranking. In *Proceedings of the Advances in Neural Information Processing Systems (NeurIPS)*. 1593–1600.

Jason Weston, Samy Bengio, and Nicolas Usunier. 2011. WSABIE: Scaling Up to Large Vocabulary Image Annotation. In *Proceedings of the Twenty-Second International Joint Conference on Artificial Intelligence (IJCAI)*. 2764–2770.

Lu Yu, Chuxu Zhang, Shangsong Liang, and Xiangliang Zhang. 2019. Multi-order Attentive Ranking Model for Sequential Recommendation. In *Proceedings of the Thirty-Third AAAI Conference on Artificial Intelligence (AAAI)*. 5709–5716.

Fajie Yuan, Guibing Guo, Joemon M. Jose, Long Chen, Haitao Yu, and Weinan Zhang. 2016. LambdaFM: Learning Optimal Ranking with Factorization Machines Using Lambda Surrogates. In *Proceedings of the 25th ACM International Conference on Information and Knowledge Management (CIKM)*. 227–236.

Weinan Zhang, Tianqi Chen, Jun Wang, and Yong Yu. 2013. Optimizing Top-n Collaborative Filtering via Dynamic Negative Item Sampling. In *Proceedings of the 36th international ACM SIGIR conference on Research and development in information retrieval (SIGIR)*. 785–788.

Tong Zhao, Julian McAuley, and Irwin King. 2014. Leveraging Social Connections to Improve Personalized Ranking for Collaborative Filtering. In *Proceedings of the 23rd ACM International Conference on Information and Knowledge Management (CIKM)*. 261–270.

