# OpenReview forum: "A Simple and Debiased Sampling Method for Personalized Ranking"
_ICLR.cc/2022/Conference — ICLR 2022 Submitted_

### Official Review · Reviewer_zgRr · 2021-11-02

**Correctness:** 4
**Technical Novelty And Significance:** 3
**Empirical Novelty And Significance:** 3
**Recommendation:** 8
**Confidence:** 4

**Main Review:**

strength:
1. This work formalizes the class-imbalance problem existing in the contrastive learning loss for recommendation task and gives a simple but effective solution.
2. The theoretical analysis provided by the authors can well support the claims.
3. Sufficient experiment results demonstrate the superiority of the proposed method.

weakness:
1. The proposed method seems to be specialized in recommendation problems.
2. The presentation can be further improved with more discussion of the main content.

Learning from class-imbalance data has been sufficiently explored during the last few years. However, most of them focus on the imbalance issue caused by the number of labels but discard the same problem existing in the contrastive learning methods. This work attempts to explore the imbalance issue when we optimize the recommender with the help of contrastive loss. We usually extract a contrastive training example with the assumption that the predicted similarity of a pair of nodes in the observed edge (positive) should have larger than the unobserved (negative) one. It's noted that the nodes can appear in both positive and negative edges, and node frequency usually follows a long-tail distribution. Because of that, there exists a significant class-imbalance bias in the contrastive learning method for the recommendation task. As far as I know, this problem has been insufficiently studied. In this work, a completed theoretical analysis is given to reveal an interesting finding that the imbalance degree has a positive relation to the item degree and can lead to the infinite norm of item embeddings, in particular those suffering from the class-imbalance problem. Along with this observation, the authors propose a debiased negative sampling method that can adaptively re-weight the loss according to the ranking position of imbalanced items. The proposed approach defines a conditional probability depending on both the given user and positive items. However, it's usually difficult to obtain an effective hard negative sample from the given probability due to the large size of the item database. In order to deal with this problem, the proposed method combines a biased random walk towards the popular items and an adaptive filtering criterion for efficiently retrieving qualified hard negative samples. Thorough experimental results demonstrate the superiority of the proposed method in real-world data. Overall, this work is clearly presented with sufficient discussion and details for further reproducibility.

concerns:
1. The proposed method seems to greedily select items with a large prediction score. While those items might be the true positive. Could you please give a justification about how to deal with the false negative issue?

2. Is the re-weighting method an essential choice? what if we remove the weight?

3. Could you provide more details on why RejectSampler depends on a biased random walk?



**Summary Of The Paper:**

This is an interesting and solid paper that theoretically demonstrates the connection between a long-tail distribution and the essentiality of hard negative sampling, proposes a simple yet effective negative sampling method, and conducts sufficient experiments.

**Summary Of The Review:**

This work gives sufficient theoretical analysis about the class-imbalance problem and provides a simple but effective debiasing method to advance the optimization performance for item recommendation tasks. Sufficient experiments are provided. Overall, it's an interesting, solid and well-organized paper studying an important problem.

---

> ### Author Response · Authors · 2021-11-21
> **Detailed response to reviewer zgRr**
>
> Dear reviewer zgRr: we are really glad to see that you appreciate this piece of work, and thank you so much for your careful comments. Below we will provide a detailed response with the hope to address the lingering points of concerns.
>
> **R2Q1:** It's a very good point to raise concern about the "false negative issue". Actually, we have a specific discussion in the Appendix E4 due to the limited space. The proposed method is not directly designed to overcome the false negative issue, but the proposed adaptive strategy can avoid to push "false negative" item far away from the users with the help of ranking-aware weight $w_{ui}$.
>
> **R2Q2:** The answer is Yes. The re-weighting method can bring several benefits. First, it can help to control the popularity bias. For those popular items, they can converge much faster than the long-tailed items. Through utilizing the ranking-aware weight $w_{ui}$, we can dynamically control the attention paid to the training samples by considering the learning performance. Second, as we reply to the Q1, the weight term can help to alleviate the false negative issue. If we remove the weight term $w_{ui}$, the ranking performance can significantly downgrade.
>
> **R2Q3:** With a given positive item $i$, the RejectSampler can randomly access to arbitrary negative item $j$, but it will decide to choose the item $j$ with acceptance probability $min${$\frac {\pi(j)} {\pi(i)}, 1$}. The selected item $j$ could also be a positive item for some users. Then, the RejectSampler can decide next item to choose according to the same transition probability. The sampling procedure actually equals to a biased random walk with a transition matrix $P^*$ as shown in the Section 4.1.

---

### Official Review · Reviewer_QcpY · 2021-11-03

**Correctness:** 2
**Technical Novelty And Significance:** 2
**Empirical Novelty And Significance:** 2
**Recommendation:** 3
**Confidence:** 5

**Main Review:**

Strengths:
1.	This paper focuses on an important and practical problem in training pairwise ranking models for recommender systems.
2.	The proposed method is simple but effective, which satisfies the requirements for a suitable negative sampling method.

Weakness:
1.	The novelty of this paper is limited. First, the analysis of the vertex-level imbalance problem is not new, which is a reformulation of the observations in previous works [Rendle and Freudenthaler, 2014; Ding et al., 2019]. Second, the designed negative sampler uses reject sampling to increase the chance of popular items, which is similar to the proposed one in PRIS [Lian et al., 2020].
2.	The paper overclaims on its ability of debiasing sampling. The “debiased” term in the paper title is confusing.
3.	The methodology detail is unclear in Sec. 4.2. The proposed design that improves sampling efficiency seems interesting but the corresponding description is hard to follow given the limited space.
4.	Space complexity of the proposed VINS should also be analyzed and compared in empirical studies, given that each (u, i) corresponds to a $buffer_{ui}$.
5.	Experiment results are not convincing enough to demonstrate the superiority of VINS on effectiveness and efficiency.
-	For effectiveness, the performance comparison in Table 1 is unfair. VINS sets different sample weights $W_{ui}$ in the training process, while most compared baselines like DNS, AOBPR, SA, PRIS set all sample weights as 1.
-	For efficiency, Table 2 should also include the theoretical analysis for contrast.


**Summary Of The Paper:**

This paper proposes to improve the negative sampling process for training pairwise ranking models in personalized recommender systems. Negative sampling plays an important role in the above application, as it can largely impact the model performance in terms of both training efficiency and recommendation accuracy.
Authors first analyze the vertex-level imbalance problem that exists in current sampling-based methods, i.e., popular items are unevenly sampled w.r.t. their chances as positive and negative. They observe that this may cause the imbalanced distribution of embedding norms between popular and long-tailed items, which can possibly harm the training efficacy and efficiency.
To cope with this problem, this paper proposes a simple but effective negative sampling method. The core idea is to choose a negative candidate with larger popularity than the given positive item. To maintain sample quality, the dynamic relative rank position of positive and negative items is also considered, which has already proven to be useful in previous works. Specifically, the proposed VINS method is achieved through a bias sampler with reject probability, which cooperates with an item buffer to enable efficient sampling of informative instances.
Empirical results on several real-world datasets demonstrate its superiority on both efficiency and effectiveness.

**Summary Of The Review:**

As mentioned in the weakness, the limitations on novelty, methodology design, and conducted experiments prohibit its acceptance.

---

> ### Author Response · Authors · 2021-11-21
> **Response to reviewer QcpY that we hope can make reviewer reconsider the assessment.**
>
> Dear Reviewer QcpY: Thanks so much for careful comments. We really hope that our response to the raised concerns can make you reconsider the assessment about this work.
>
> R2Q1: this work can be regarded as a complementary study by explicitly formalizing the vertex-imbalance problem and its impact with a theoretical analysis. As a pilot study about the impact of long-tail item distribution, [Rendle et al. WSDM 2014] presented an empirical analysis about the evolution of gradient scale along with the observation that popular items can be easily attributed with a large prediction score. After checking the mention papers [Rendle et al. WSDM 2014, Ding et al. IJCAI 2019], we find that they did not explicitly present the vertex-imbalance issue by considering the two-side of each item, and neither realize that the prediction value for popular items might just be biased by the their embedding norm.
>
> Most of previous samplers including the mentioned method (PRIS) tend to reject a random sampled item $j$ if $e_{uj}\in E$, while select it as the negative item with probability 1 if $e_{uj}\not \in E$. While RejectSampler rejects the sampled candidate by comparing its weight with the target positive item $i$. However, it’s not fair to compare the RejectSampler with the mentioned method, because RejectSampler just works as a component of the proposed method VINS. From perspective of reject sampling method, VINS contains a two-phase rejection setting. It not only rejects the sampled positive items, but also set another criterion to filter out negative item by comparing the prediction difference between the sampled items with the positive item. While PRIS just reject sampled positive items.
>
> R2Q2 and Q3: Sorry about the confused terms. Through the theoretical analysis, we find that performance discrepancy can easily happen between popular and long-tailed items. In particular, those popular items tend to have larger prediction, while the unpopular items converge much slower with relative small gradients. It can be regarded as a kind of bias caused by the item popularity. In this work, we leverage the idea of inverse propensity weight to control the bias through estimating a ranking-aware weight $w_{ui}$. With this term, we can use the knowledge from the model itself to better balance the attention paid to the training samples. Moreover, we can avoid push away the potential false negative items hiding in the sampled hard negative samples with a relative small gradient. Most of hard negative sampling method discard the essentiality of avoiding to push the false negative away from the anchor points. For the section 4.2, we will reorganize the draft and make the presentation about the methodology in a concise way. We have a pseudocode in Appendix.C to present the complete algorithm with more details.
>
> R2Q4: the space complexity for the buffer can be implemented in an efficient way with only complexity O(1). Actually we only need to store the sample with the maximum prediction score. More details can be found from Line 4-13 in Algorithm 1 of Appendix.C.
>
> R2Q5: in terms of the fairness comparing with the baselines, we follow the exact methodology given by the original work. For the mentioned baselines like DNS, AOBPR, PRIS, the design principles behind them does not pay attention to the impact of the bias caused by the item popularity. But it's interesting to study what kind of things can happen after integrating the weight term. We conduct experiments to valid their performance in Steam and Amazon CDs&Vinyl data. The results shown in Table 1 and 2 for baselines are presented in the form of x/y, where x stands for the results obtained after modifying the objective function, and y stands for the original results. We can see that applying the weight $w_{ui}$ to DNS can improve the ranking performance, while the performance of AOBPR and PRIS downgrade a little bit.
>
> | Method | F1@10 | NDCG@10 |
> | :- | :-: | :-: |
> | AOBPR  | 0.0283/0.0334 | 0.0396/0.0463|
> | PRIS  | 0.0357/0.0374 | 0.0469/0.0502  |
> | DNS  | 0.0407/0.0398 | 0.0568/0.0551  |
> | VINS | 0.0429 | 0.0594 |
>
> Table 1. Experiment results to show the performance of baselines by considering the dynamic weight strategy in Steam data.
>
> |  Method   | F1@10  | NDCG@10|
> | :- | :-: | :-: |
> | AOBPR  | 0.0236/0.0211 | 0.0310/0.0278|
> | PRIS  | 0.0245/0.0252 | 0.0324/0.0331  |
> | DNS  | 0.0295/0.0275 | 0.0392/0.0366  |
> | VINS | 0.0310 | 0.0410 |
>
> Table 2. Experiment results to show the performance of baselines by considering the dynamic weight strategy in Amazon CDs&Vinyl data.
>
> In terms of the time complexity, we will give a theoretical comparison about the time complexity with the baselines in the revised manuscript. Thank so very much for pointing this issue.

---

> > ### Comment · Reviewer_QcpY · 2021-11-28
> > **Response**
> >
> > Thanks for the above detailed response.
> >
> > > From perspective of reject sampling method, VINS contains a two-phase rejection setting. It not only rejects the sampled positive items, but also set another criterion to filter out negative item by comparing the prediction difference between the sampled items with the positive item. While PRIS just reject sampled positive items.
> >
> > For Q1, the proposed two-phase rejection setting seems incremental compared with previous works.
> >
> > > the space complexity for the buffer can be implemented in an efficient way with only complexity O(1). Actually we only need to store the sample with the maximum prediction score. More details can be found from Line 4-13 in Algorithm 1 of Appendix.C.
> >
> > For Q4, according to the original paper,
> >
> > > We use an item buffer $buffer_{uj}\in V_i$ with size $\kappa$  to store every sampled negative candidate $j$
> >
> > > For VINS, we need to search the best choice for buffer size $\kappa$ and decay factor $\beta$. In this work, we ﬁnd that $\kappa=64$ or $128$ is good enough according to the analysis results.
> >
> > It seems to me that buffer is designed for each $(u,i)$ pair, which can be huge in practical applications considering the number of user-item interactions.
> >
> > For Q5, thank you for adding this extra experiment. However, negative sampling and sample reweighting are generally two mutual alternative approaches for handling the problem of missing instances in personalized ranking. My remaining concern is that the proposed method seems more complicated as it requires both two parts。

---

> > > ### Author Response · Authors · 2021-11-30
> > > **Further discussion about the raised concerns**
> > >
> > > Dear QcpY:
> > >
> > > Thank you so much for reading our response and giving a further feedback to let us know your concerns.
> > >
> > > 1. Both of VINS and PRIS employ the reject sampling strategy. However, they have fundamental difference from each other. PRIS assumes that an item to be sampled as a negative one follows distribution $P( j | u)$. While VINS follows a conditional sampling distribution $P(j | u,i)$ to reject the sampled items. For more information, you can refer to the Appendix E2, where we have a elaborated discussion about the connection to previous works.
> > >
> > > 2. The term "buffer" might cause some misunderstanding. The buffer works as a constraint to limit the maximum steps to find a violated negative sample. We do not need store every sampled item. Theoretically, the space complexity for each $(u,i)$ is $O(1)$. In practical applications, we can reuse the variable of negative item $j$ in the training triplet $(u,i,j)$ for the buffer to store the sampled negative candidate which has the maximum prediction but still not violates the criterion yet. Therefore, the space complexity will not increase too much, even for large-scale recommendation task.
> > >
> > > 3. Last but not least, the way to obtain the weight $w_{ui}$ is closely associated with the negative sampling procedure. The weight can be regarded as the additional product. Without the adaptive sampling method, it would be very difficult for us to estimate the weight. The proposed method focuses on advancing the quality of negative sampling method. The weight can be obtained by using the memorized steps to find a violated negative sample.
> > >
> > > Thanks again for your careful comments. We really hope that our response can help to further address the lingering point of concerns and reconsider the assessment about the work. If you have any concern, please do not hesitate to let us know.

---

### Official Review · Reviewer_A9Ak · 2021-11-05

**Correctness:** 4
**Technical Novelty And Significance:** 2
**Empirical Novelty And Significance:** 1
**Recommendation:** 3
**Confidence:** 4

**Main Review:**

To overcome vertex-level imbalance issue, the authors propose a two-phase sampling approach dubbed as Vital Negative Sampler (VINS), by sampling a negative item with a larger degree (to explicit positive connection between item degree and the learned embeddings) and close prediction score to the given positive sample (to leverage relative rank position of positive and negative items for finding more informative negative samples).


The paper is well-written and the authors supported the claims made with theoretical and empirical analysis. However there few key issues that prevents me from giving it a high-score and makes me to lean toward rejection:

(1) While the mentioned issues totally make sense and it is appreciated that authors conducted experiments to provide intuition, but in objective (1) the regularization terms are dropped. Focusing on Imbalanced Item Theorem that indicates the embedding of popular items becomes extremely large, but I was left wondering if this issue can not be alleviated via regularization. Note that the number of times the embedding of an item  will be regularized is proportional to the number of times it is sampled, which possibly avoids embedding explosion as claimed. Even more interesting, if we adaptively pick the regularization parameter for each item proportional to its degree, it might resolve the issue.

(2) While the proposed two-stage sampling idea looks interesting, in my opinion it looks incremental and lacks enough novelty. For example this is simply the hard negative sampling in contrastive learning or even utilizing a dynamic weighted loss based on the loss of current model on each negative sample (note that here we can sample a batch of negative samples and avoid the computation burden of VINS).

(3) The proposed idea can be considered as a sampling variant of ranking with accuracy at top (push-norm) where we try to push as many as possible negative samples below positive items by changing the pairwise ranking loss into a push-norm based loss. It would be interesting to discuss the proposed idea in context of this line of research

Rudin, C. (2009). The P-Norm Push: A Simple Convex Ranking Algorithm that Concentrates at the Top of the List. Journal of Machine Learning Research, 10, 2233-2271.

Li, Nan, Rong Jin, and Zhi-Hua Zhou. "Top Rank Optimization in Linear Time." NIPS. 2014.

**Summary Of The Paper:**

This paper investigates personalized ranking from implicit feedback where we leverage a set of positively labeled data (for which interactions exist such as clicking, purchase etc) and use all the items with no explicit feedback as negative instances and aim to learn a ranking model for recommendation. The idea is to minimize a loss over triplets (u,i, j) over users, a positively labeled instance and a negatively labeled instance to learn the embeddings for items and users.

The main observation in this paper is that in addition to known unbalancedness issue on edge level (proportion of positive instance to negative instance), the existing methods also suffer from vertex-level imbalance problem due to fact that due to sampling, the number of times an item appears as positive and negative is disproportionate. In particular,  popular items with degree (positiveness) greater than average item degree are under-sampled as negative samples, while cold-start items with degree less than average degree are over-sampled as negative instances. This in turn makes the norm of learned embeddings for popular items towards infinite after a certain training iterations. This issue even occurs if one utilizes an under-sampling/oversampling to solve the edge-level imbalance issue. The observations are supported both theoretically and empirically.


**Summary Of The Review:**

Overall, I like the observations made and the proposed sampling idea, but due to issues discussed above I found the contribution incremental and leaning towards rejection.

---

> ### Author Response · Authors · 2021-11-21
> **Restatement of our motivation and contribution that we hope can help to reconsider the assessment.**
>
> Dear reviewer A9Ak, we really appreciate your careful comments and sincerely hope that our response can help to address the lingering points of concern. Below we provide a detailed response to the mentioned issues.
>
> **Motivation:** Learning a personalized ranking model with contrastive loss has its unique challenges. One of the most difficult thing is how to keep a balance between the false negative and hard negative samples. Typically we will regard the items that are interacted by users as the positive, while the other items as the negative. However, we might overlook the existence of false negative samples because of the oversimplified assumption on the items that are not interacted by the users. Usually it's very difficult to tell the false negative from the hard negative samples without clear labels. Bot of them might be very close to the anchor points in the representation space. Because of this challenge, it deserves a special attention to design a flexible hard negative sampling method which adjust the sampling probability according to the useful knowledge drawn from the model itself.
>
> **Response to Q1:** The study of class-imbalance issue aim at providing important clues for designing effective and efficient sampling method, but not for simply controlling the norm of item embeddings. Actually we follow the idea of BPR [3] and l2-norm regularization is added to the objective function. Regularization technology works well to drop penalty on the model complexity, but not good at finding the useful negative samples.
>
> **Response to Q2 and Q3:** Thanks for your kind suggestion. The mentioned research [1,2] provide pioneering attempts to formalize the ranking optimization by considering the hard examples. Such kind of idea is actually shared by the research field of recommender systems. However, it's still challenging to directly apply them to personalized recommendation task.
>
> 1. Scalibility: generally, the size of item set can be millions or even billions. While pioneering studies like [1,2] are based on several assumptions that might hold for small-scale data, but not for personalized recommendation problems with large scale of items. It's impossible to compare every pair of positive and negative samples due to the sparsity of user behaviors. The enormous number of items can bring significant challenges to the accessibility to the hardest examples. Taking the TopPush [2] as example, the number of dual variables can be $|U|\times|I|$ and the derived solution based on a linear model is not applicable to optimizing the deep learning model.
>
> 2. False Negative: In recommendation task, the interactions between users and items can be formalized as bipartite network. The missing edges does not exactly mean that a user dislike the items. Because of this issue, we have to deal with the false negative samples when designing the negative samplers. Moreover, each item can be a positive or negative sample for different users, which is different from the scenario [1,2] with clean contrastive samples.
>
> It's true that the proposed method is an alternative sampler that searches the hard negative samples. The problem that we are studying contains lots of uncertainty. When trying to find the hard negative samples, we need to avoid hurting the false negative ones. In this study, we find that those positive items will have inconsistent learning performance. Some of them, especially those popular items, will be pushed close to the top very quickly, while the unpopular grows much slower. It's a kind of bias caused by the item popularity. In this work, we control the bias through estimating a ranking-aware weight $w(r)$, which allows the contrastive loss to carefully deal with the negative items with close predicted scores as the positive items. Most of hard negative sampling method discard the essentiality of avoiding to push the false negative away from the anchor points. The weight $w(r)$ can help to alleviate this problem by controlling how far to push the negative samples by measuring its closeness to the prediction of positive items. In the case of sampling a batch of negative samples, many pioneering works already put this idea into practice, such as the baselines SA, PRIS, DNS. Most of them greedily give a large weight to the negative samples proportion to their prediction score, while missing the uncertainty hiding in the user preferences. The experimental results could further provide some clues to support the importance of considering the uncertainty.
>
> References:
> 1. Rudin, C. (2009). The P-Norm Push: A Simple Convex Ranking Algorithm that Concentrates at the Top of the List. Journal of Machine Learning Research, 10, 2233-2271.
>
> 2. Li, Nan, Rong Jin, and Zhi-Hua Zhou. "Top Rank Optimization in Linear Time." NIPS. 2014.
>
> 3. Rendle et al. BPR: Bayesian Personalized Ranking from Implicit Feedback, UAI 2009.

---

### Official Review · Reviewer_1zJH · 2021-11-08

**Correctness:** 3
**Technical Novelty And Significance:** 2
**Empirical Novelty And Significance:** 3
**Recommendation:** 6
**Confidence:** 3

**Main Review:**

1. the paper studies an important topic on negative sampling for the learning-to-rank problem
2. the proposed method is based on an intuitive analysis on the vertex level imbalance sampling instead of commonly used edge level sampling.
3. It provided both theoretical analysis and experiments on real world datasets to demonstrate the effectiveness on the proposed sampling approach.

**Summary Of The Paper:**

For pair-wise learning-to-rank problems, it often suffers from class imbalance problem when constructing training data where negative class examples are much more frequent then positive class examples. Negative example sampling is an important differentiator on training and classifier performance. this paper proposed a method on negative class sampling by considering the degree of vertexes. It proposed a rejection function in order to sample with higher probability from high degree vertex instead of using edge level random sampling. It provided both theoretical analysis and experiments on real world datasets to demonstrate the effectiveness on the proposed sampling approach.

**Summary Of The Review:**

The paper is marginally innovative. The method it proposed is intuitive and clear. However, it is unclear if the proposed methods is applicable to any form of learning-to-rank problems or it is tied to certain learning-to-rank models, such as MF. It would be very useful to clarify the how generic the proposed sampling approach is applicable explicitly. For the learn-2-rank models mentioned in the paper,  I would recommend it for a poster paper, the writing can also be improved by directly citing the models used in the experiments instead of listing them in appendix.

---

> ### Author Response · Authors · 2021-11-21
> **Response to Reviewer 1zJH**
>
> Dear reviewer 1zJH: We really thank for your careful comments. I'm very pleased that you confirm the contributions of our simple and effective method for learning-to-rank problem as well as the our experimental results. The main concern from the reviewer is about the whether the proposed method is limited to the specific models. Though we take factorization models like MF as a base model, the proposed method employs a model-agnostic scoring function to dynamically reject the easy but select the hard negative sample for pairwise learning problem. Therefore, it is applicable to any kind of models. We will update the draft with an explicit statement. Thank you so much for your suggestion to further improve the presentation of this work.

---

### Decision · Program_Chairs · 2022-01-20

**Decision:**

Reject

**Comment:**

The reviewers remained concerned about the overall novelty of the paper, finding the contributions somewhat incremental. The authors are encouraged to better substantiate design choices that they make, to improve the overall presentation, and to contrast with the works/line of research brought up by the reviewers.